# Prevalence and diagnostic reliability of *BRAF, RAS* mutations, and *RET/PTC* rearrangements in a Latin American public health service population with thyroid nodular disease

Freddy David Moposita Molina[1], Grasiela Agnes[2], Marília Remuzzi Zandoná[2], Rogério Izquierdo[3], Fábio Alves Bilhar[4], Laura Berton Eidt[5], Virgílio Gonzales Zanella[6], Luiz Felipe Osowski[6], Sofia de Oliveira Belardinelli[2], Amanda Cometti de Andrade[2], Erika Laurini de Souza Meyer[5,7], Lenara Golbert[5,7], Vanessa Suñé Mattevi[1,2,8]*

**1** Graduate Program in Biosciences, Federal University of Health Sciences of Porto Alegre (UFCSPA), Brazil, **2** Molecular Biology Laboratory, UFCSPA, Brazil, **3** Radiology Unit, Santa Casa de Porto Alegre, Brazil, **4** Endocrine Unit, Hospital Nossa Senhora da Conceição, Grupo Hospitalar Conceição, Brazil, **5** Endocrine Unit, Santa Casa de Porto Alegre, Brazil, **6** Head and Neck Unit, Santa Casa de Porto Alegre, Brazil, **7** Department of Internal Medicine, UFCSPA, Brazil, **8** Department of Basic Health Sciences, UFCSPA, Brazil

* vmattevi@ufcspa.edu.br

## Abstract

Despite their high prevalence and generally benign nature in most cases, the investigation of thyroid nodules still presents potential diagnostic pitfalls, especially in cases with indeterminate cytology results. The performance of molecular markers of thyroid cancer may vary across centers and populations. This study aimed to verify the prevalence of mutations in the *BRAF,* and *RAS* genes, and *RET/PTC* rearrangements in patients undergoing fine-needle aspiration biopsy (FNAB) for thyroid nodule evaluation in a real-world public health service population. Point mutations and rearrangements were detected by Sanger DNA sequencing. A total of 231 thyroid nodules in 220 patients were evaluated, being 86.8% females and a mean age of 55.6 ± 13.9 years. For molecular analysis, high-quality DNA and RNA were obtained from 200 samples. Mutations or rearrangements in target genes were identified in 14% of the 200 samples evaluated. The frequency of the *BRAF*-like mutations was 5.5%, detected in 9 out of 17 malignant nodules (52.9%) and one in a benign nodule (0.7%). Fourteen *RAS*-like mutations were identified in benign nodules (57.1% *HRAS*, 21.5% *NRAS* and 21.5% *KRAS*) and only one was present in a malignant nodule (5.9%). Considering only nodules with indeterminate cytology (Bethesda III and IV, n = 53), 9 mutations were detected, 6 in benign histology (all *RAS*-like), 1 in malignant histology (*BRAF*-like), and 2 still unoperated, therefore without a histopathological diagnosis. This research concludes that the presence of the *BRAF* V600E mutation could be useful in supporting the diagnosis of thyroid cancer, due to its high positive predictive value, since 89% of nodules with *BRAF* V600E mutation

**Data availability statement:** Data cannot be shared publicly because they refer to patients personal information. Data are available only if authorized by the Institutional Ethics Committee (contact via cep@santacasa.tche.br) for researchers who meet the criteria for access to confidential data.

**Funding:** CNPQ - Edital Universal 2018 UFCSPA Edital PROPPG nº 44/2024 The funders had no role in study design, data collection and analysis, decision to publish, or preparation of the manuscript.

**Competing interests:** The authors have declared that no competing interests exist.

were malignant. Additionally, clinical criteria should be established to determine which nodules with *RAS*-like mutations require closer follow-up, particularly those with indeterminate cytology.

## Introduction

Thyroid cancer is the most common endocrine malignancy, with an estimated 43,800 new cases diagnosed in the United States in 2022, making it the seventh most frequent carcinoma among women [1], presenting typically as a thyroid nodule. Over 60% of the population develops at least one thyroid nodule as they age [2]. However, only 5% to 15% of these nodules contain malignant thyroid neoplasms. Traditionally, the diagnosis of thyroid nodules has relied on ultrasound and cytological evaluation. Nevertheless, these methods have limitations, especially in nodules with indeterminate cytology, where the risk of malignancy can vary significantly [2,3].

About 25% of thyroid nodules aspirated thorough ultrasound-guided fine-needle aspiration biopsy (US-FNAB) are classified as atypia of undetermined significance (AUS) (Bethesda III) and follicular neoplasm (FN) (Bethesda IV), where cancer cannot be ruled out with certainty [4–7]. Within these categories, the prevalence shows significant variability among studies, with a risk of malignancy ranging from 13-30% for AUS and 23–34% for FN [5,8]. Surgery not only involves a high economic cost, but it also carries risks of morbidity, such as damage to nearby structures and the requirement of thyroid hormone replacement therapy throughout life [9–11]. Additionally, it is essential to note that approximately 75% of nodules initially classified as indeterminate are found to be benign after histopathological evaluation [5,9,12].

At the molecular level, thyroid cancer frequently harbors genetic alterations that activate the MAPK and PI3K/AKT signaling pathways, which are critical for tumorigenesis and progression [13]. Molecular analyses have allowed the identification of genetic mutations and expression patterns that improve diagnostic accuracy. Among the most relevant molecular markers are *BRAF* V600E mutation and *RET/PTC*1 and *RET/PTC*3 rearrangements, which have been widely studied and validated in different studies [14–17]. In these cases, the presence of specific mutations may guide toward more aggressive management or, on the contrary, avoid unnecessary surgeries in patients with low-risk lesions [18,19]. Nevertheless, the most common mutations in indeterminate nodules are mutations in the *RAS* genes (*NRAS* and *HRAS* codon 61, *KRAS* codons 12/13). These are considered weaker driver mutations associated with a lower positive predictive value (PPV). Recent studies have shown that the PPV of *RAS* mutations varies between 10% and 37% [20–22].

Thyroid cancer is a multifactorial phenotype, and Latin American and African populations have been significantly underrepresented in existing studies of complex diseases [23]. Their inclusion is of paramount importance for the complete comprehension of complex phenotypes, because it is known that global differences in the prevalence and distribution of complex diseases and their risk factors are the result of

a combination of demographic, environmental, and genetic factors. The prevalence of molecular alterations for indeterminate nodules is poorly described in Latin America [24].

In this context, the present study focuses on determining the prevalence of the main mutations in the *BRAF* and *RAS* genes and the *RET/PTC*1 and *RET/PTC*3 rearrangements in patients with thyroid nodular disease in a South Brazilian public health system setting to evaluate their diagnostic utility in real-world contexts.

## Materials and methods

### Patients

This is a prospective diagnostic accuracy study conducted in accordance with the STARD (Standards for Reporting of Diagnostic Accuracy Studies) guidelines [25]. Consecutive adult patients, aged ≥ 18 years, with one or more thyroid nodule(s) who underwent FNA under ultrasonographic guidance (FNA-US), or patients with indeterminate cytology who were scheduled for diagnostic surgery between 2019 and 2024, were recruited from the Endocrinology Division of Santa Casa de Porto Alegre and Hospital Nossa Senhora da Conceição, two tertiary public hospitals in the South of Brazil. All patients underwent a complete clinical evaluation and thyroid ultrasonography. Patients with known thyroid cancer were excluded. The study followed the principles of the Declaration of Helsinki and was approved by the local ethics committees, with all participants providing written informed consent (CAAE ISCMPA: 53201915.0.0000.5345 and HNSC: 29368320.0.3002.5530).

### US examination

Thyroid Ultrasound Conventional B-mode and Doppler images of the neck and thyroid gland were obtained by ultrasound machine using a high-frequency probe (12 MHz). All US examinations from Santa Casa were performed by the same radiologist (RI) and the same endocrinologist at HNSC (FAB), both with more than 15 years of experience in thyroid ultrasound. The number, size, and features of the thyroid nodules were recorded, and each nodule was classified according to the ACR-TIRADS ultrasound risk-stratification system. The presence of cervical lymphadenopathy was also evaluated.

### Thyroid FNA, sample collection, cytology and histology

FNA-US was performed with a disposable needle (21G) connected to a 10 ml disposable syringe. All patients submitted to FNA-US for one or more nodules ≥1 cm according to the ACR-TIRADS and/or other clinical indication criteria were included for this study.

In most cases, two to three FNA passes were made. Most of the aspirated sample from the first two passes was used to prepare direct cytological smears, while the residual material from the new layer and the needle wash from both passes were placed in a tube containing 500 µl of RNAlater™ (Life Technologies, Carlsbad, CA) nucleic acid preservative solution, where it was stored until genetic material extraction for no longer than 3 months.

Pathologists experienced in thyroid cytopathology interpreted aspiration specimens, and results were reported according to the six categories of the 2023 Bethesda System for Reporting Cytopathology: I) non-diagnostic; II) benign; III) atypia of undetermined significance (AUS); IV) follicular neoplasm (FN); V) suspicious for malignancy; and VI) malignant [5].

FNA thyroid samples from indeterminate nodules (Bethesda III and IV) scheduled for surgery were collected at Santa Casa de Porto Alegre during surgery by the head and neck surgeons. Surgery was indicated by the assistant team in Bethesda III (in accordance with clinical judgment) and IV nodules. Bethesda II nodules were also indicated for surgery when there were associated compressive symptoms and/or thyroid autonomy. Surgical pathology specimens obtained from thyroidectomy were reviewed and interpreted by a staff pathologist, according to the World Health Organization Guidelines from 2022 [19]. Histopathology for each nodule was recorded. The reference standard used for this study was cytology and/or histopathology: for benign nodules, Bethesda II and, for malignant nodules, Bethesda VI; the histopathological diagnosis results of indeterminate thyroid nodules obtained through surgery were also used when available.

## Molecular analysis

**Nucleic acid isolation.** FNA samples and surgical biopsies were analyzed at the Molecular Biology Laboratory of the Federal University of Health Sciences of Porto Alegre (UFCSPA). Total DNA and RNA were extracted using the Trizol protocol (TRI-pure isolation, Roche), as described by Kizys (2012) [26]. The concentration and quality of the extracted nucleic acids were assessed using the BioSpecNano spectrophotometer (Shimadzu, Japan) to ensure optimal downstream applications.

**Assessment of sample adequacy for molecular testing.** RNA samples were evaluated to determine the proportion of thyroid epithelial cells in samples to ensure that a minimum amount (10%) of these cells were present. To this end, the expression of the cytokeratin 7 (*KRT7*) gene was evaluated, which is specific to certain epithelial cell types, including thyroid epithelial cells. *GAPDH* gene expression was used as a constitutive control. The retrotranscription of total RNA to complementary DNA (cDNA) was performed according to the manufacturer's protocol (random hexamers plus oligonucleotides dT and reverse transcriptase enzyme) (High-Capacity cDNA Reverse Transcription Kit, Thermo Scientific, Waltham, Massachusetts).

Gene amplification was accomplished via quantitative real-time polymerase chain reaction (qPCR) with the detection of the amplified products with hydrolysis probes (TaqMan) in a StepOne Plus Real-Time System (Life Technologies, Waltham, Massachusetts). A sample was deemed representative of thyroid tissue if the Ct value difference between the *GAPDH* and *KRT7* genes was less than 3.5 ($Ct_{KRT7} - Ct_{GAPDH} < 3.5$), thereby indicating that a minimum of 10% of the cells within the sample were of thyroid origin [3].

**Mutation research.** Point mutations *BRAF* V600E, codon 61 of *NRAS*, codon 61 of *HRAS*, and codons 12 and 13 of *KRAS* were detected by DNA analysis using amplification by PCR with primers described in Table 1 and Sanger sequencing with Big Dye reagent in a SeqStudio Genetic Analyzer (Thermo Fisher Scientific Waltham, Massachusetts) [3]. Subsequently, nodules positive for the *BRAF* V600E mutation were selected for *TERT* promoter mutation analysis. For *RET/PTC1* and *RET/PTC3* rearrangements, primers were designed to flank the fusion point (Table 1), and the cDNA synthesized in the previous step was used as input for PCR amplification. The products of the gene fusions were observed through agarose gel electrophoresis stained with ethidium bromide. Positive results were confirmed by Sanger sequencing of the PCR products.

## Statistical analysis

Clinical, laboratory, ultrasonography, and cytological data are reported as mean ± standard deviation (SD) values, as median with range (for asymmetrical continuous variables), or as absolute numbers and percentages (categorical variables). Data were compared between benign and malignant nodules using the Mann-Whitney U-test or chi-squared test, as appropriate.

Cytology and histopathological examination served as the reference standard for determining malignancy. In instances of Bethesda II cytology, follicular adenomas (FA), follicular nodular thyroid disease (TFND), and lymphocytic thyroiditis (LT), a negative molecular test result was deemed as a true negative, whereas a positive result was regarded as a false positive. Conversely, if the histopathological diagnosis revealed carcinoma, a negative molecular test result was categorized as a false negative.

To determine sensitivity (S), specificity (E), positive predictive value (PPV), negative predictive value (NPV), positive likelihood ratio (LR+), negative likelihood ratio (LR−), and diagnostic accuracy, the calculations were performed as follows: S = true positives/ (true positives + false negatives); E = true negatives/ (true negatives + false positives); PPV = true positives/ (true positives + false positives); NPV = true negatives/ (true negatives + false negatives); LR+ = sensitivity/ (1 − specificity); LR− = (1 − sensitivity)/ specificity; diagnostic accuracy = (true positives + true negatives)/ (true positives + true negatives + false positives + false negatives).

The data obtained were analyzed using the statistical software SPSS version 20.0 (IBM, Armonk, NY).

**Table 1. Primers list for PCR.**

| Gene | Primer | Primers list for PCR (5' to 3') | Amplicon size |
|---|---|---|---|
| *BRAF (codon 600)* | | | |
| | Forward | TCATAATGCTTGCTCTGATAGGA | |
| | Reverse | GGCCAAAAATTTAATCAGTGGA | 229 bp |
| *HRAS (codon 61)* | | | |
| | Forward | GTCCTCCTGCAGGATTCCTA | |
| | Reverse | ATGGCAAACACACACAGGAA | 152 bp |
| *NRAS (codon61)* | | | |
| | Forward | CCCCTTACCCTCCACACC | |
| | Reverse | TGGCAAATACACAGAGGAAGC | 161 bp |
| *KRAS (codon 12/13)* | | | |
| | Forward | AAGGCCTGCTGAAAATGACTG | |
| | Reverse | GGTCCTGCACCAGTAATATGCA | 165 bp |
| *TERT (codon 228/250)* | | | |
| | Forward | CACCCGTCCTGCCCCTTCACCTT | |
| | Reverse | GGCTTCCCACGTGCGCAGCAGGA | 193 pb |
| *RET/PTC1* | | | |
| | Forward | TGCAGCAAGAGAACAAGGTG | |
| | Reverse | TTTCAGATGGAAGGCCGTTG | 199 bp |
| *RET/PTC3* | | | |
| | Forward | CCTTTCAGCGAATGGCTCCT | |
| | Reverse | ATTCCCACTTTGGATCCTCCT | 136 bp |

PCR, polymerase chain reaction

## Results

### Clinical, ultrasonographic and cytology evaluation

During the study period, 220 patients submitted to FNA-US were included. Of these, 86.8% were women; the mean age was 55.6±13.9 years (Table 2). Two or more thyroid nodules were found in 70.5% of patients, and 5.5% had a family history of thyroid cancer. The nodules evaluated had diameter ranging from 0.7 to 14.8 cm, with a median of 3.1 cm.

Of the nodules evaluated, 26% (n=60) were classified as indeterminate, with 13.9% in category III and 12.1% in category IV (Table 2). Of these, 30 were collected through ultrasound-guided FNA (FNA-US) and 30 samples were obtained during surgery (FNA-surgical).

In Table 3, patients were compared according to the presence of *BRAF-like* mutations, *RAS-like* mutations, or absence of mutation. Statistically significant differences were observed in age at diagnosis, with mutation-negative patients being older (mean 57.2 years) compared to the *BRAF-like* and *RAS-like* groups (means of 46.2 and 45.3 years, respectively). In terms of ultrasound features, all nodules with *BRAF-like* mutations were classified as moderate and highly suspicious (ACR TIRADS 4–5), while 35.3% of nodules with *RAS-like* mutations were benign to low suspicious (ACR TIRADS 1–3). The nodules in the *BRAF-like* group were significantly smaller (median 1.9 cm) compared to the no-mutation (median 3.1 cm) and RAS-like (median 2.9 cm) groups. No significant differences were observed regarding gender, family history of thyroid cancer, or other neoplasms.

### Analysis of mutations in the total sample

From the 231 nodules initially collected through FNA-US, 31 samples had insufficient nucleic acids, or the proportion of epithelial cells was less than 10% (CtKRT7 – CtGAPDH>3.5). Therefore, quality DNA and RNA were obtained from 200 samples (86.6%) (Fig 1).

**Table 2. Demographic and clinical-pathological characteristics of patients with thyroid nodules.**

| Parameters | Patients (n = 220) |
|---|---|
| **Gender: n (%)** | |
| **Female** | 191 (86.8) |
| **Age at diagnosis (years)** | |
| Mean SD | 55.6 ± 13.9 |
| **Two or more nodules: n (%)** | 155 (70.5) |
| **History with other malignancies** | 39 (17.7) |
| **Exposure to cervical radiation therapy** | 5 (2.3) |
| **Family history of thyroid cancer** | 12 (5.5) |
| **Parameters** | **Nodules (n = 231)** |
| **Nodule diameter (cm)** | |
| Range | 0.7 - 14.8 |
| Median | 3.1 |
| **Ecographic score: n (%)** | |
| ACR-TIRADS 1/2/3 | 71 (30.7) |
| ACR-TIRADS 4 | 119 (51.5) |
| ACR-TIRADS 5 | 41 (17.8) |
| **Cytology: n (%)** | |
| Bethesda I | 11 (4.8) |
| Bethesda II | 141 (61.0) |
| Bethesda III | 32 (13.9) |
| Bethesda IV | 28 (12.1) |
| Bethesda V | 10 (4.3) |
| Bethesda VI | 9 (3.9) |
| **Nodules undergoing thyroidectomy** | 69 (29.9) |
| **Malignant nodules** | 18 (7.8) |

Molecular analysis revealed 28 mutations, representing 14% of the total samples analyzed, of which 11 were *BRAF-like* (10 *BRAF* V600E and one *RET/PTC1* fusion) and 17 were *RAS-like* (9 *HRAS* codon 61, 5 *NRAS* codon 61, and 3 *KRAS* codons 12/13). The results of the cytology/histology and detected mutations are shown in Fig 1.

Of the 17 nodules with *RAS-like* mutations, the distribution according to the Bethesda category was as follows: Bethesda II (6), Bethesda III (3), Bethesda IV (4), and Bethesda V (4). Of these, only one was malignant, corresponding to a Bethesda V nodule with an *NRAS* mutation. On the other hand, the 11 nodules with *BRAF-like* mutations were distributed as follows: Bethesda II (1), Bethesda III (1), Bethesda IV (1), Bethesda V (1), and Bethesda VI (7). All these nodules were malignant, except for the nodule classified as Bethesda II. This case underwent total thyroidectomy due to goiter with compressive symptoms, and no malignant lesions were detected in two re-evaluations by an expert thyroid pathologist.

TERT promoter mutation analysis was initially planned for all 10 nodules harboring the *BRAF* V600E mutation. However, due to sample quality limitations, only eight nodules were successfully analyzed. None of these showed mutations in the *TERT* promoter.

### Relationship between cytology, histopathology and molecular evaluation

**Samples with benign cytology (Bethesda II).** The majority of analyzed samples came from the Bethesda II classification, with 123 nodules analyzed. Seven samples showed mutations (5.7%), distributed as follows: 6 *RAS* and

**Table 3. Distribution of demographic, clinical, and ultrasound parameters according to mutation profile.**

| Parameters | No mutation (n = 172) | BRAF-like (n = 11) | RAS-like (n = 17) | p |
|---|---|---|---|---|
| **Gender: n (%)** | | | | 0.280* |
| Male | 19 (11.0) | 3 (27.3) | 2 (11.8) | |
| Female | 153 (89.0) | 8 (72.7) | 15 (88.2) | |
| **Age at diagnosis (years)** | | | | **0.002**** |
| Mean SD | 56.8 ± 14.6 | 46.3 ± 14.5 | 45.4 ± 16.6 | |
| Range | 20 - 82 | 28–68 | 18 - 83 | |
| **Nodule diameter (cm)** | | | | 0.051** |
| Median | 3.2 | 1.9 | 2.9 | |
| Range | 0.7 - 14.8 | 1.3 - 4.7 | 0.9 - 9 | |
| **Ecographic score: n (%)** | | | | 0.084* |
| ACR-TIRADS 1/3 | 52 (30.2) | 0 (0) | 6 (35.3) | |
| ACR-TIRADS 4/5 | 120 (69.8) | 11 (100) | 11 (64.7) | |
| **Family history of thyroid cancer: n (%)** | | | | 0.860* |
| No | 163 (94.8) | 1 (9.1) | 1 (5.9) | |
| Yes | 9 (5.2) | 10 (90.9) | 16 (94.1) | |
| **Family history of other neoplasms: n (%)** | | | | 0.281* |
| No | 148 (86.0) | 8 (72.7) | 16 (94.1) | |
| Yes | 24 (14.0) | 3 (27.3) | 1 (5.9) | |

**Kruskal-Wallis test;

*By X2 test. In bold significant p values (<0.05).

1 *BRAF* V600E. Of these, two were referred for thyroidectomy. One of the six *RAS-like* mutations was associated with TFND (thyroid follicular nodular disease), as was the nodule with the *BRAF* V600E mutation. The other five cases were not surgically treated (Fig 2).

No malignancy cases were observed for nodules classified as Bethesda II (negative cytology), regardless of the presence of mutations (Table 4).

**Samples with indeterminate cytology (III and IV).** This group included 53 nodules with indeterminate cytology. Of them, 9 samples (16.9%) with mutations were identified, distributed as follows: 2 *BRAF* V600E and 7 *RAS-like* (3 *NRAS*, 2 *HRAS*, and 2 *KRAS*) (Fig 1). No *RET/PTC* fusion was detected. Of the two nodules with the *BRAF* V600E mutation, one was diagnosed as papillary thyroid carcinoma (PTC) and classified in the Bethesda IV category, while the second nodule (Bethesda III) is still awaiting surgery. Meanwhile, the 7 nodules with *RAS-like* mutations were post-surgically diagnosed as follows: four with follicular adenoma (FA), two with TFND, and one is still awaiting surgery (Fig 2).

The prevalence of malignancies in indeterminate categories was 18.2% for Bethesda III and 20.0% for Bethesda IV (Fig 3). As shown in Table 4, nodules with indeterminate cytology had a 19% probability of malignancy based on cytology alone. The specificity and sensitivity for *BRAF*-like mutations were 100% and 17%, respectively (Fig 3). In contrast, nodules positive for *RAS*-like mutations were not associated with malignancy.

Additional diagnostic performance parameters for the combined *BRAF* and *RAS* mutation group: LR+=0.71, LR-= 1.09, and overall diagnostic accuracy = 64.5%.

**Samples with cytology suspect of malignant and malignant (Bethesda V and VI).** In the 10 nodules from Bethesda V category, five mutations were identified, four of which were *RAS-like* and diagnosed post-surgically as follows: two TFND, one PTC, and one sample still awaiting surgery. The sample with the *BRAF* V600E mutation was diagnosed as PTC (Fig 1).

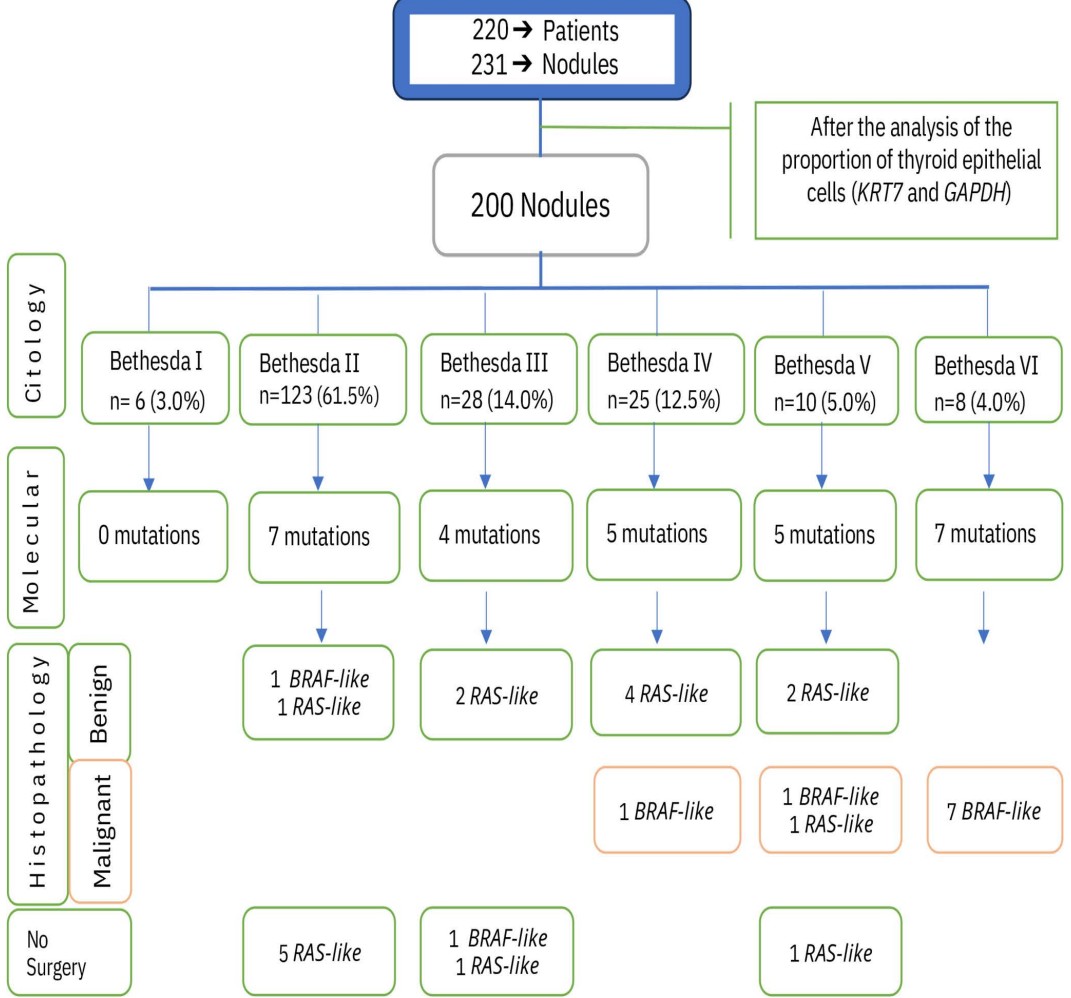

**Fig 1. Classification of thyroid nodules based on cytological analysis.** The diagram illustrates the classification of thyroid nodules according to their cytology based on the 2023 Bethesda system, the presence of mutations and the subsequent post-surgery histopathological results [5,19].

Eight FNA samples with a cytological diagnosis of Bethesda VI were analyzed. Seven (87.5%) had mutations: six with *BRAF* V600E and one with the *RET/PTC1* fusion. Five of the six *BRAF* V600E mutations were associated with Classic PTC, and one with the infiltrative follicular variant PTC (IFV-PTC). The nodule with the *RET/PTC1* fusion was also diagnosed as IFV-PTC. In contrast, the nodule without a detected mutation was classified as PTC. It is important to note that the *BRAF* V600E mutation was present in 75% of nodules with a Bethesda VI classification.

## Histological analysis

Upon analyzing 200 nodules with a molecular diagnosis, we observed that 56 samples underwent thyroidectomy. The cytological, histological, and mutation results are presented in Fig 2. Of these, 39 were confirmed as benign through histology, while 17 were classified as malignant.

Upon examining these 39 benign nodules, we found that 11 of them, classified as Bethesda II, were operated on due to compressive symptoms or hyperthyroidism. In this subgroup, mutations were detected in two cases: one with a *BRAF*

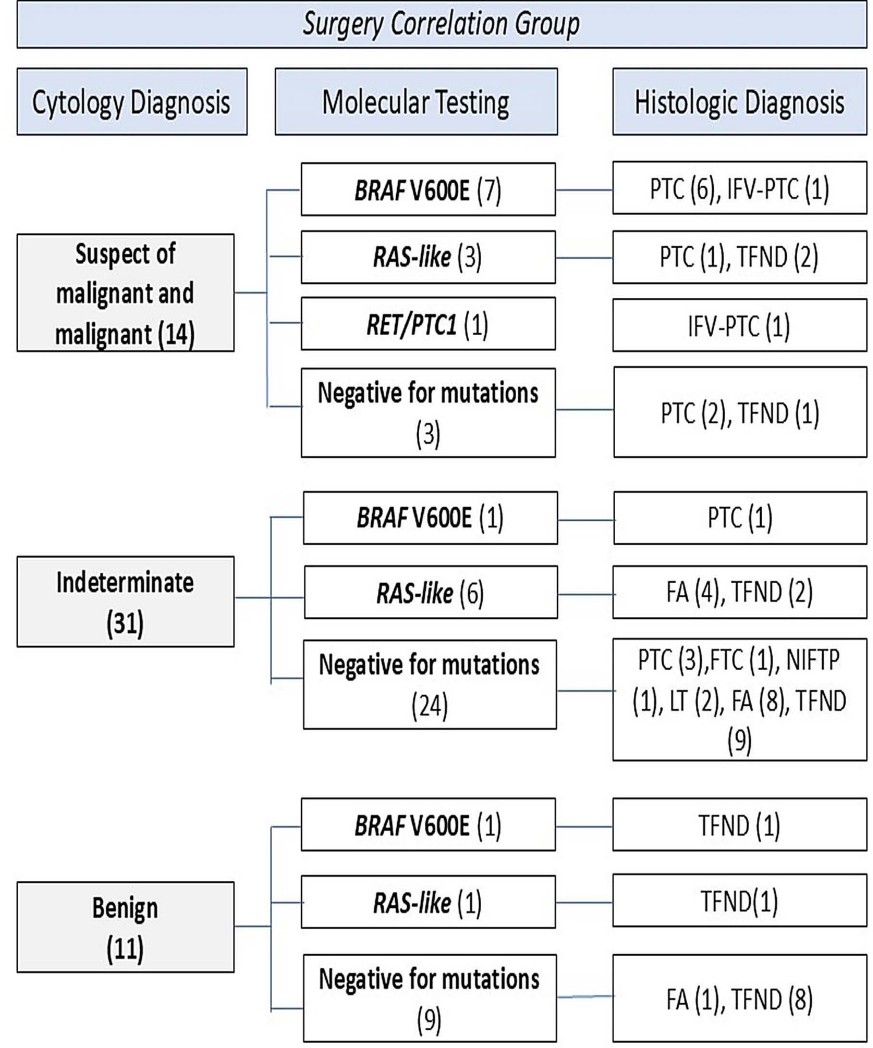

**Fig 2. Correlation between cytology, molecular findings, and histological diagnosis.** LT: lymphocytic thyroiditis; TFND: thyroid follicular nodular disease; FA: follicular adenoma; PTC: papillary thyroid cancer; IFV-PTC: infiltrative follicular variant papillary thyroid carcinoma; FTC: follicular thyroid cancer; FNA: fine-needle aspiration; NIFTP: non-invasive follicular thyroid neoplasm with papillary-like nuclear features [5,19].

**Table 4. Probability of cancer in thyroid nodules depending on the results of cytological and molecular analysis.**

| Results of cytology and molecular analysis | Cancer probability (%) |
|---|---|
| Positive cytology (Bethesda V and VI) and positive for mutation | 82 |
| Indeterminate cytology and positive for *BRAF*-like | 100 |
| Indeterminate cytology and positive for *RAS*-like | 0 |
| Indeterminate cytology and negative for mutation | 21 |
| Indeterminate cytology | 19 |
| Negative cytology (Bethesda II) and negative for mutation | 0 |
| Negative cytology (Bethesda II) | 0 |

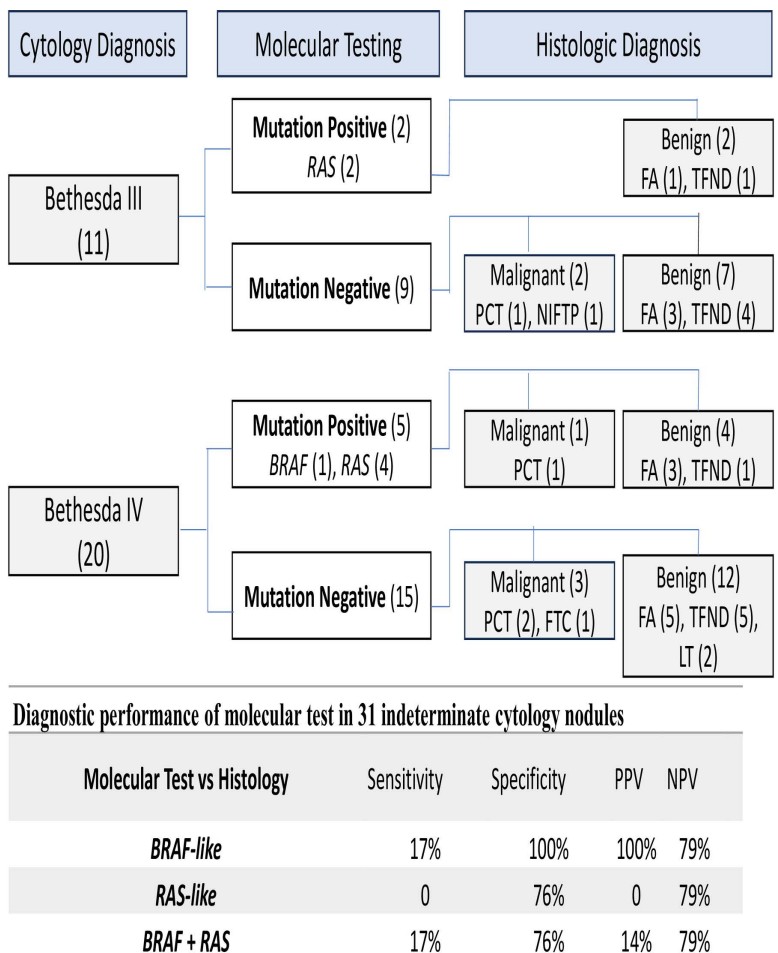

## Diagnostic performance of molecular test in 31 indeterminate cytology nodules

| Molecular Test vs Histology | Sensitivity | Specificity | PPV | NPV |
|---|---|---|---|---|
| *BRAF-like* | 17% | 100% | 100% | 79% |
| *RAS-like* | 0 | 76% | 0 | 79% |
| *BRAF + RAS* | 17% | 76% | 14% | 79% |

PPV: positive predictive value; NPV: Negative predictive value

**Fig 3. Performance of molecular testing in specific categories of indeterminate FNA cytology.** NIFTP (Noninvasive Follicular Thyroid Neoplasm with Papillary-like Nuclear Features) was considered a malignant neoplasm [5,19].

mutation and another with a *RAS* mutation, both diagnosed as TFND. Meanwhile, the remaining 28 benign nodules, 25 with indeterminate cytology and 3 within the Bethesda V category, had mutations in 8 cases, all *RAS-like* type.

Considering the total number of malignant nodules, we found that 58.8% (10 out of 17) had mutations, including 8 PTC and 2 IFV-PTC. Of these mutations, 9 were *BRAF-like* and 1 was *RAS-like*: 7 in Bethesda VI, 2 in Bethesda V, and 1 in Bethesda IV (Fig 1). The nodules without mutations included 5 PTC, 1 FTC, and 1 NIFTP (Fig 2). Notably, the prevalence of cancer associated with *BRAF-like* mutations was 52.9%, while for *RAS-like* mutations it was only 5.9%.

## Discussion

Molecular panels have increasingly been employed in the evaluation of thyroid nodules to optimize diagnostic accuracy and minimize the need for unnecessary interventions. However, for the correct use of these panels, it is necessary to know the universe of existing mutations in a population, as well as the malignancy rates for each cytological category (pre-test probability). To this end, we prospectively evaluated 200 patients with thyroid nodules and investigated the prevalence of the main mutations described as predictors of malignancy in a Latin American population.

Apart from age and nodule size, no significant differences were observed in other demographic or clinical characteristics among patients with and without mutations. The smaller size of *BRAF-like* nodules may reflect earlier detection, as these mutations are often linked to highly suspicious ultrasound features [27]. A statistically significant difference in age at diagnosis was observed between patients with and without mutations, with mutation-negative individuals being significantly older than those harboring *BRAF-like* or *RAS-like* mutations. While several studies have reported a higher prevalence of mutations, particularly *BRAF* V600E in older patients and their association with more aggressive tumor behavior [28], the relationship between age and mutation status has not been consistently demonstrated across different cohorts. These discrepancies may reflect variations in population genetics, environmental exposures, or referral and diagnostic practices. In contrast to the association of *BRAF-like* mutations with high-risk ultrasonographic scores, the investigation of indeterminate nodules harboring *RAS-like* mutations presents challenges due to their often ambiguous ultrasonographic, cytological, and molecular profiles, which frequently position them within a diagnostic gray zone.

In our sample, with a cancer prevalence of 7.8%, multigene panel analysis revealed a mutation rate of 14%. *BRAF-like* mutations accounted for 39% of positive samples, while the remaining 61% corresponded to *RAS-like* mutations. Among the nodules that underwent thyroidectomy (n = 56), we found that 58.8% of malignant nodules exhibited a genetic mutation, with a predominance of 90% *BRAF-like* mutations (*BRAF* V600E and *RET/PTC*). It is worth highlighting that the highest number of mutations in malignant nodules was found in those with Bethesda VI cytology (70%), while the remaining 30% were within Bethesda IV and V categories. In these cases, the presence of the *BRAF* V600E mutation could be helpful in supporting the diagnosis, given their high positive predictive value (PPV), as 89% of nodules with this mutation were malignant.

Although no *TERT* promoter mutations were found among the eight *BRAF* V600E-positive nodules in our series, several larger studies have reported co-occurrence rates ranging from approximately 5% to 16% [29,30]. The absence of *TERT* alterations in our small series likely reflects the limited sample size.

The prevalence of *RAS-like* mutations may vary according to the type of nodule, patient age, and other factors. In indeterminate thyroid nodules, *NRAS* mutations are typically the most frequent, followed by *KRAS* and *HRAS* [9,20,31]. In our sample, from the South of Brazil population, *NRAS* mutations (5.6%) were also the most common in indeterminate nodules, followed by *HRAS* and *KRAS* (both 3.7%). However, when considering the total sample, *HRAS* mutations (4.5%) were the most frequent, followed by *NRAS* (2.5%) and *KRAS* (1.5%). Among these, *NRAS* was the only mutation detected in a PTC associated with malignancy. These findings suggest a possible regional variation in the distribution of these mutations, highlighting the importance of our research in describing mutational variations within the Brazilian population.

A considerable number of *RAS-like* mutations (9.3%) were identified in benign nodules, including non-operated nodules classified as Bethesda II, underscoring the need for patient monitoring to assess the risk of progression. It is important to highlight that *RAS* is a potent oncogene, and there is some indirect evidence suggesting that it may promote malignant transformation and tumor dedifferentiation in thyroid cells [9,19,32]. However, whether benign nodules with *RAS* mutations can progress to follicular carcinomas or exhibit more aggressive behavior is still unclear [32]. Given their frequent presence in benign nodules, *RAS-like* mutations should be interpreted with caution in cases of indeterminate cytology. In this context, active surveillance may be a reasonable approach, especially when no additional clinical or ultrasound risk factors are present.

Notably, 76% of benign nodules that underwent surgery and had a preoperative indeterminate classification did not present mutations. This result aligns with existing research indicating that most indeterminate nodules, between 82% and 94%, are not malignant and do not require surgical intervention [3,5,20,33].

It's noteworthy that the pretest probability of malignancy was 19% for the indeterminate nodules, similar to that described in the Bethesda Classification [5]. The low sensitivity in indeterminate nodules is due to the low frequency of mutations associated with malignancy. Furthermore, mutations detected in benign nodules reduced the test's specificity. These factors limit the panel's accuracy in differentiating benign from malignant lesions. As a result, the negative likelihood

ratio (LR-) exceeds 1, demonstrating that the test has limited usefulness for ruling out malignancy in indeterminate nodules. When analyzing *BRAF-like* and *RAS-like* mutations separately, we found that the isolated presence of *BRAF*-like mutations appears to have a high PPV of 100%, showing an association with thyroid cancer, specifically papillary thyroid carcinoma (PTC). In contrast, *RAS*-like mutations were associated with a low PPV. Studies such as that by Guan et al. in 2020, report that mutations in the *RAS* gene have a low PPV, ranging from 10% to 37% [20–22]. Additionally, most reports indicate that *RAS-like* mutations are associated with follicular carcinomas (FTC) [9,12,32]; however, in our study, only one case of FTC was presented.

Although the overall mutation detection rate was relatively low (14%), the main impact of our study was to describe the prevalence of mutations in consecutive nodules, particularly highlighting the high frequency of benign nodules with *RAS-like* mutations. The clinical utility was most evident in *BRAF-like* mutations, which confirmed malignancy. Future studies with larger samples, particularly with more indeterminate nodules, are needed to better evaluate the accuracy and clinical utility of molecular testing. To support this, we are actively expanding our sample to include more indeterminate and malignant nodules, aiming to strengthen our findings and validate these preliminary results.

This study has certain limitations, the limited number of nodules with indeterminate cytology included, the patients not yet having undergone surgeries and the low prevalence of FTC and other follicular patterns lesions could have influenced our results, specifically the applicability of our evaluation in indeterminate cytology. Additionally, 41.2% of malignant nodules tested negative, exceeding the 10%−30% reported in other studies [5,14,15], possibly due to unstudied mutations beyond our panel's scope.

Evidently, the use of Sanger sequencing does not allow searching for new mutations in target genes located outside the primer-delimited regions. However, its high accuracy and low cost make it more suitable for limited-resource settings, which are the reality in most Global South countries. For instance, our test has been developed at a relatively low cost, estimated at just US$8 per examination, making it more accessible to a real-world population served by the Public Health System. This cost is significantly lower than the US$1,500–4,000 typically charged for commercial tests in the US [34].

In conclusion, the high PPV associated with *BRAF*-like mutations, along with a relatively low cost, could serve as a useful tool for diagnosing thyroid cancer. Further studies are needed to determine the possible mutations that were not detected by our molecular panel. Moreover, a set of clinical criteria needs to be identified to determine which nodules with *RAS*-like mutations require closer follow-up, particularly those with indeterminate cytology.

## Author contributions

**Conceptualization:** Erika Laurini de Souza Meyer, Lenara Golbert, Vanessa Suñé Mattevi.

**Data curation:** Freddy David Moposita Molina, Grasiela Agnes, Marília Remuzzi Zandoná, Laura Berton Eidt, Virgílio Gonzales Zanella, Luiz Felipe Osowski, Sofia de Oliveira Belardinelli, Amanda Cometti de Andrade, Erika Laurini de Souza Meyer, Lenara Golbert, Vanessa Suñé Mattevi.

**Formal analysis:** Freddy David Moposita Molina, Lenara Golbert, Vanessa Suñé Mattevi.

**Funding acquisition:** Lenara Golbert.

**Investigation:** Freddy David Moposita Molina, Rogério Izquierdo, Fábio Alves Bilhar, Laura Berton Eidt, Erika Laurini de Souza Meyer.

**Methodology:** Freddy David Moposita Molina, Grasiela Agnes, Marília Remuzzi Zandoná, Rogério Izquierdo, Fábio Alves Bilhar, Virgílio Gonzales Zanella, Luiz Felipe Osowski, Sofia de Oliveira Belardinelli, Amanda Cometti de Andrade.

**Project administration:** Lenara Golbert.

**Resources:** Lenara Golbert.

**Supervision:** Erika Laurini de Souza Meyer, Lenara Golbert, Vanessa Suñé Mattevi.

**Writing – original draft:** Freddy David Moposita Molina, Lenara Golbert, Vanessa Suñé Mattevi.

**Writing – review & editing:** Erika Laurini de Souza Meyer, Lenara Golbert, Vanessa Suñé Mattevi.

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
