## [Decision Letter · Decision Letter 0]

Dear Dr. Suñé Mattevi,

Thank you for submitting your manuscript to PLOS ONE. After careful consideration, we feel that it has merit but does not fully meet PLOS ONE’s publication criteria as it currently stands. Therefore, we invite you to submit a revised version of the manuscript that addresses the points raised during the review process.

**The manuscript addresses an interesting and important clinical problem,  which is the the potential of molecular markers to a pre-diagnostic evaluations. Nevertheless, there are several questions raised by the reviewers which need to be  answered point by point, and that will improve the manuscript quality.**

We look forward to receiving your revised manuscript.

Kind regards,

Paula Boaventura, PhD

Academic Editor

PLOS ONE

**Journal Requirements:**

1. When submitting your revision, we need you to address these additional requirements. Please ensure that your manuscript meets PLOS ONE's style requirements, including those for file naming. The PLOS ONE style templates can be found at https://journals.plos.org/plosone/s/file?id=wjVg/PLOSOne_formatting_sample_main_body.pdf and https://journals.plos.org/plosone/s/file?id=ba62/PLOSOne_formatting_sample_title_authors_affiliations.pdf 2. Thank you for stating the following financial disclosure: CNPQ - Edital Universal 2018UFCSPA Edital PROPPG nº 44/2024  Please state what role the funders took in the study.  If the funders had no role, please state: "The funders had no role in study design, data collection and analysis, decision to publish, or preparation of the manuscript." If this statement is not correct you must amend it as needed. Please include this amended Role of Funder statement in your cover letter; we will change the online submission form on your behalf. 3. We note that you have included the phrase “data not shown” in your manuscript. Unfortunately, this does not meet our data sharing requirements. PLOS does not permit references to inaccessible data. We require that authors provide all relevant data within the paper, Supporting Information files, or in an acceptable, public repository. Please add a citation to support this phrase or upload the data that corresponds with these findings to a stable repository (such as Figshare or Dryad) and provide and URLs, DOIs, or accession numbers that may be used to access these data. Or, if the data are not a core part of the research being presented in your study, we ask that you remove the phrase that refers to these data.

Reviewers' comments:

Reviewer's Responses to Questions

**Comments to the Author**

1. Is the manuscript technically sound, and do the data support the conclusions?

Reviewer #1: Yes

Reviewer #2: Yes

Reviewer #3: Partly

2. Has the statistical analysis been performed appropriately and rigorously?

Reviewer #1: Yes

Reviewer #2: No

Reviewer #3: I Don't Know

3. Have the authors made all data underlying the findings in their manuscript fully available?

Reviewer #1: Yes

Reviewer #2: No

Reviewer #3: Yes

4. Is the manuscript presented in an intelligible fashion and written in standard English?

Reviewer #1: Yes

Reviewer #2: Yes

Reviewer #3: Yes

**Reviewer #1:**  General comment:

Many patients with thyroid nodules are subjected to unnecessary surgery, which often results in comorbidities and high costs for the healthcare system. This manuscript (PONE-D-25-10645) poses a good question, regarding the potential use of molecular markers to a pre-diagnostic evaluation. This approach would help distinguish malignant and/or aggressive nodules from those that are indolent and can be safely monitored rather than surgically removed.

Strong points of the study:

• Real-world public health service population from two institutions from a Brazilian/Latin American population.

• Appropriate statistical analysis that considered parametric and non-parametric tests when applicable.

• Usage of the most recent WHO Thyroid nodule Classification guidelines (2022).

• Usage of the most recent Bethesda System for Reporting Thyroid Cytopathology (2023).

• Ethical issues were addressed, and ethical committee approval was obtained.

• Patients were recently diagnosed (2019-2024), which results in better quality tissue samples.

• Patients with already know thyroid cancer were excluded, not to be confounding variables.

• Patients were analyzed by the same radiologist and endocrinologist, ensuring consistency in the evaluation and characterization.

Weak points of the study:

The manuscript presents good data, but the authors do not go very far in the statistical analysis. It would have been interesting to have analyzed the clinicopathological characteristics of the patients regarding their molecular status, ecographic score, cytology results and so on. Despite the manuscript being focused on the molecular marker’s diagnostic value, the available data could be more explored.

Given to the fact that this is a real-world public health service population, which is great to mimic everyday clinical practice, it is expected that the number of malignant cases is low, therefore, it makes it more difficult to demonstrate the author’s point of view. The low number of malignant tumors and of mutations makes it hard to take major conclusions. Is there a possibility to increase the number of cases studied? What is the next step for this work?

Minor revisions:

• The authors do not explain how the positive and negative predictive values were calculated. Please add that information to the methodology section and in the table of Figure 3.

• In the Figures, please add the corresponding citations of the WHO and Bethesda classifications.

• Table footnotes should have indications of the statistical tests/calculations used in each evaluation.

• Financing entities are not well described or to whom that financing was attributed. Please clarify.

• Some section titles in the manuscript are rather brief and could benefit from greater specificity.

• Was the BRAF V600E+ benign nodule subjected to re-evaluation? Is it possible that this case is actually a malignant lesion? According to the 2022 WHO, benign lesions and low-risk neoplasms do not present with BRAF mutation.

• Despite using the 2022 WHO classification, some tumors are classified as FVPTC. However, this category is no longer present in this classification. FVPTC tumors must be classified as Invasive encapsulated follicular variant papillary thyroid carcinoma (IEFV-PTC) or infiltrative follicular variant papillary thyroid carcinoma (IFV-PTC). Please correct in the text and in the Figures.

• Page 3, line 33 – typo: unintended paragraph.

• Page 5, line 75 – What does the nucleic acid preservative solution consist of, and how long were the samples stored in it?

• Page 5, line 87 – I suggest indicating in the sentence that the WHO guidelines utilized were those from 2022.

• Page 9, line 187 – typo: instead of “malignant”, it is written “malingnant”.

• Figure 3 – NIFTP is classified as malignant, but it is however a low-risk neoplasm. NIFTP does not fall in the benign nor in the malignant category. Please change this in the Figure or create a footnote explaining this.

• Quantitative real-time PCR is written differently in the manuscript (RT-PCR and qPCR).

General comment:

In general, I found this manuscript interesting and easy to follow. It poses an interesting hypothesis, and the conclusions are supported by the presented data. The limitations of the study were acknowledged by the authors and are justified by inherent characteristics of the studied population.

The manuscript could benefit from a deeper statistical analysis between clinicopathological data the reported parameters.

For the BRAF mutated tumors, I would suggest the authors to add the analysis of TERTp mutations to the panels, for these have been shown to have a strong predictive value for tumors that are more aggressive and metastatic. Especially, when in the present of BRAF mutation as well. Please see these studies: https://doi.org/10.3390/cancers13092048 ; https://doi.org/10.1530/EDM-23-0025.

**Reviewer #2: ** This is a well-written article based on a clinically useful study. Following observations require authors` attention:

a. The study design is mainly "Diagnostic Accuracy" but authors have not mentioned this design anywhere in this manuscript.

b. Checklist for Diagnostic Accuracy studies, available at : (http://www.equator-network.org/reporting-guidelines/stard) should be followed and mentioned in the manuscript.

c. Some parameters like "Likelihood Ratios" and overall "Diagnostic Accuracy of genetic tests should be done on the already collected data.

d. Result shows very high specificity (100%) and very low sensitivity but these findings have neither been elaborated in discussion nor mentioned in conclusion.

**Reviewer #3: ** Article interesting as it address the important issue in nodules, raise many possible questions:

Critiques:

1. How does the relatively low overall mutation detection rate (14%) impact the clinical utility of molecular testing in this population?

2.Was there any correlation between the mutation status and specific ultrasound features or risk stratification systems (e.g., TIRADS)?

3. How were benign vs. malignant histology determined for nodules that were not surgically removed or had inconclusive follow-up data (e.g., the two indeterminate cytology cases without histopathology)?

4Given that RAS-like mutations were more frequently found in benign nodules, how should clinicians interpret these mutations in the context of indeterminate cytology?

5.How might the use of Sanger sequencing and RT-PCR, rather than more sensitive techniques like NGS, have limited the detection of low-frequency or novel mutations?

6. Thyroid cancer frequently harbor PI3K/AKT and MAPK pathway alteration resulting in activation of these pathways. This should be indicated by citing, Pubmed ID: 31689493; Pubmed ID: 27387551.

**Do you want your identity to be public for this peer review?** For information about this choice, including consent withdrawal, please see our Privacy Policy

Reviewer #1: No

Reviewer #2: **Yes: ** Prof Aamir Ijaz

Reviewer #3: No

---

## [Author Response · Author response to Decision Letter 1]

28 Jun 2025

Response to the Reviewer's comments:

# Journal Requirements:

CNPQ - Edital Universal 2018

UFCSPA Edital PROPPG nº 44/2024

Response: The funders had no role in study design, data collection and analysis, decision to publish, or preparation of the manuscript. Included in the cover letter.

Response: the phrase was removed.

Reviewer #1:

1 - Weak points of the study:

The manuscript presents good data, but the authors do not go very far in the statistical analysis. It would have been interesting to have analyzed the clinicopathological characteristics of the patients regarding their molecular status, ecographic score, cytology results and so on. Despite the manuscript being focused on the molecular marker’s diagnostic value, the available data could be more explored.

Response: As requested, more information regarding demographic and clinical-pathological features in patients with thyroid nodules were included in Table 3 and in the Results section, lines 255-265 and in the discussion section, lines 383-394.

2 - Given to the fact that this is a real-world public health service population, which is great to mimic everyday clinical practice, it is expected that the number of malignant cases is low, therefore, it makes it more difficult to demonstrate the author’s point of view. The low number of malignant tumors and of mutations makes it hard to take major conclusions. Is there a possibility to increase the number of cases studied? What is the next step for this work?

Response: We agree with the reviewer. Although it is not possible to increase the numbers of cases presented herein, we are actively expanding our sample to include more indeterminate and malignant nodules, aiming to strengthen our findings and validate these preliminary results. These information were included in the manuscript, lines 455-463.

3-Minor revisions:

• The authors do not explain how the positive and negative predictive values were calculated. Please add that information to the methodology section and in the table of Figure 3.

Response: The information were included in the Methodology section, lines 223-230.

To determine sensitivity (S), specificity (E), positive predictive value (PPV), negative predictive value (NPV), positive likelihood ratio (LR+), negative likelihood ratio (LR−), and diagnostic accuracy, the calculations were performed as follows: S = true positives / (true positives + false negatives); E = true negatives / (true negatives + false positives); PPV = true positives / (true positives + false positives); NPV = true negatives / (true negatives + false negatives); LR+ = sensitivity / (1 − specificity); LR− = (1 − sensitivity) / specificity; diagnostic accuracy = (true positives + true negatives) / (true positives + true negatives + false positives + false negatives).

• In the Figures, please add the corresponding citations of the WHO and Bethesda classifications.

Response: The citations were included in the legend of Fig. 1 (line 276), Fig. 2 (line 313), and Fig. 3 (line 337).

• Table footnotes should have indications of the statistical tests calculations used in each evaluation.

Response: included (line 252).

• Financing entities are not well described or to whom that financing was attributed. Please clarify.

Response: The financing entities were:

CNPQ (Conselho Nacional de Desenvolvimento Científico e Tecnológico; National Council for Scientific and Technological Development, Brazil) - Edital Universal 2018

UFCSPA (Universidade Federal de Ciências da Saúde de Porto Alegre, Federal University of Health Sciences of Porto Alegre, Brazil) - Edital PROPPG nº 44/2024

• Some section titles in the manuscript are rather brief and could benefit from greater specificity.

Response: The section titles were expanded: line 319: Samples with indeterminate cytology (III and IV), and line 342: Samples with cytology of suspected malignancy and malignant (Bethesda V and VI)

• Was the BRAF V600E+ benign nodule subjected to re-evaluation? Is it possible that this case lesion is actually a malignant? According to the 2022 WHO, benign lesions and low-risk neoplasms do not present with BRAF mutation.

Response: An expert thyroid pathologist of our institution, re-evaluated two times the anatomopathological exam of this case of goiter and BRAF mutation. The patient was submitted to total thyroidectomy and has no malignant lesion. Of interest, this patient has an aggressive breast cancer, with distant metastasis and is under treatment in our hospital, with no detection of thyroid cancer or cervical lesions. These information were included in lines 287 - 290.

• Despite using the 2022 WHO classification, some tumors are classified as FVPTC. However, this category is no longer present in this classification. FVPTC tumors must be classified as Invasive encapsulated follicular variant papillary thyroid carcinoma (IEFV-PTC) or infiltrative follicular variant papillary thyroid carcinoma (IFV-PTC). Please correct in the text and in the Figures.

Response: these changes were made in lines 310, 351, 352, and 368.

• Page 3, line 33 – typo: unintended paragraph.

Response: corrected

• Page 5, line 75 – What does the nucleic acid preservative solution consist of, and how long were the samples stored in it?

Response: the information were included in lines 143-145: RNAlaterTM (Life Technologies, Carlsbad, CA), where it was stored until genetic material extraction, for no longer than 3 months.

• Page 5, line 87 – I suggest indicating in the sentence that the WHO guidelines utilized were those from 2022.

Response: included in line 158 .

• Page 9, line 187 – typo: instead of “malignant”, it is written “malingnant”.

Response: corrected.

• Figure 3 – NIFTP is classified as malignant, but it is however a low-risk neoplasm. NIFTP does not fall in the benign nor in the malignant category. Please change this in the Figure or create a footnote explaining this.

Response: this information was included in the Figure footnote, lines 336 - 337.

• Quantitative real-time PCR is written differently in the manuscript (RT-PCR and qPCR).

Response: corrected.

General comment:

In general, I found this manuscript interesting and easy to follow. It poses an interesting hypothesis, and the conclusions are supported by the presented data. The limitations of the study were acknowledged by the authors and are justified by inherent characteristics of the studied population.

The manuscript could benefit from a deeper statistical analysis between clinicopathological data the reported parameters.

For the BRAF mutated tumors, I would suggest the authors to add the analysis of TERTp mutations to the panels, for these have been shown to have a strong predictive value for tumors that are more aggressive and metastatic. Especially, when in the present of BRAF mutation as well. Please see these studies: https://doi.org/10.3390/cancers13092048 ; https://doi.org/10.1530/EDM-23-0025.

Response: as requested, TERTp mutations were analyzed and these results were included in lines 199-200: “Subsequently, nodules positive for the BRAF V600E mutation were selected for TERT promoter mutation analysis.”

Lines 292-295: “TERT promoter mutation analysis was initially planned for all 10 nodules harboring the BRAF V600E mutation. However, due to sample quality limitations, only eight nodules were successfully analyzed. None of these showed mutations in the TERT promoter.”

Lines 409-412: “Although no TERT promoter mutations were found among the eight BRAF V600E-positive nodules in our series, several larger studies have reported co-occurrence rates ranging from approximately 5% to 16% [29, 30]. The absence of TERT alterations in our small series likely reflects the limited sample size. “

Reviewer #2:

This is a well-written article based on a clinically useful study. Following observations require authors` attention:

a. The study design is mainly "Diagnostic Accuracy" but authors have not mentioned this design anywhere in this manuscript.

b. Checklist for Diagnostic Accuracy studies, available at : (http://www.equator-network.org/reporting-guidelines/stard) should be followed and mentioned in the manuscript.

Response: as requested, these information were included in lines 112-113.

c. Some parameters like "Likelihood Ratios" and overall "Diagnostic Accuracy of genetic tests should be done on the already collected data.

Response: The parameters were calculated and the information was included in Statistical analysis section (lines 223 - 230), Results (339-340), and Discussion (438-444).

d. Result shows very high specificity (100%) and very low sensitivity but these findings have neither been elaborated in discussion nor mentioned in conclusion.

These information were included in lines 438-444.

“The low sensitivity in indeterminate nodules is due to the low frequency of mutations associated with malignancy. Furthermore, mutations detected in benign nodules reduced the test’s specificity. These factors limit the panel’s accuracy in differentiating benign from malignant lesions. As a result, the negative likelihood ratio (LR-) exceeds 1, demonstrating that the test has limited usefulness for ruling out malignancy in indeterminate nodules”

Reviewer #3: Article interesting as it address the important issue in nodules, raise many possible questions:

Critiques:

1. How does the relatively low overall mutation detection rate (14%) impact the clinical utility of molecular testing in this population?

Response:

Although the overall mutation detection rate was relatively low (14%), the main impact of our study was to describe the prevalence of mutations in consecutive nodules, particularly highlighting the high frequency of benign nodules with RAS-like mutations. The clinical utility was most evident in BRAF-like mutations, which confirmed malignancy. Future studies with larger samples, particularly with more indeterminate nodules, are needed to better evaluate the accuracy and clinical utility of molecular testing.

These information were included in lines 452-460.

2. Was there any correlation between the mutation status and specific ultrasound features or risk stratification systems (e.g., TIRADS)?

Response: As requested, more information regarding demographic and clinical-pathological features in patients with thyroid nodules were included in Table 3 and in the Results section, lines 250-265 and in the discussion section, lines 383-398.

3. How were benign vs. malignant histology determined for nodules that were not surgically removed or had inconclusive follow-up data (e.g., the two indeterminate cytology cases without histopathology)?

Response: Bethesda II and VI were considered as benign and malignant, respectively. Only indeterminate nodules that underwent thyroidectomy were considered in the histological analysis. Of nodules submitted to surgery, 39 were confirmed as benign through histology, while 17 were classified as malignant. These information are presented in lines 357-360.

4. Given that RAS-like mutations were more frequently found in benign nodules, how should clinicians interpret these mutations in the context of indeterminate cytology?

Response: Given their frequent presence in benign nodules, RAS-like mutations should be interpreted with caution in cases of indeterminate cytology. In this context, active surveillance may be a reasonable approach, especially when no additional clinical or ultrasound risk factors are present. These information were included in the Discussion section, lines 429-432.

5.How might the use of Sanger sequencing and RT-PCR, rather than more sensitive techniques like NGS, have limited the detection of low-frequency or novel mutations?

Response: Evidently, the use of Sanger sequencing does not allow searching for new mutations in target genes located outside the primer-delimited regions. However, its high accuracy and low cost make it more suitable for limited-resource settings, which are the reality in most Global South countries. For instance, our test has been developed at a relatively low cost, estimated at just US$8 per examination, making it more accessible to a real-world population served by the Public Health System. This cost is significantly lower than the US$1,500–4,000 typically charged for commercial tests in the US. These information were included in the Discussion section, lines 468-474.

6. Thyroid cancer frequently harbor PI3K/AKT and MAPK pathway alteration resulting in activation of these pathways. This should be indicated by citing, Pubmed ID: 31689493; Pubmed ID: 27387551.

Response: The requested information was included in the Introduction section, lines 83-85, as follows: At the molecular level, thyroid cancer frequently harbors genetic alterations that activate the MAPK and PI3K/AKT signaling pathways, which are critical for tumorigenesis and progression.

---

## [Decision Letter · Decision Letter 1]

Prevalence and Diagnostic Reliability of BRAF, RAS Mutations, and RET/PTC Rearrangements in a Latin American Public Health Service Population with Thyroid Nodular Disease

PONE-D-25-10645R1

Dear Dr. Suñé Mattevi,

We’re pleased to inform you that your manuscript has been judged scientifically suitable for publication and will be formally accepted for publication once it meets all outstanding technical requirements.

Kind regards,

Paula Boaventura, PhD

Academic Editor

PLOS ONE

Additional Editor Comments (optional):

Reviewers' comments:

Reviewer's Responses to Questions

**Comments to the Author**

Reviewer #1: All comments have been addressed

Reviewer #3: All comments have been addressed

2. Is the manuscript technically sound, and do the data support the conclusions?

Reviewer #1: Yes

Reviewer #3: Yes

3. Has the statistical analysis been performed appropriately and rigorously?

Reviewer #1: Yes

Reviewer #3: I Don't Know

4. Have the authors made all data underlying the findings in their manuscript fully available?

Reviewer #1: Yes

Reviewer #3: Yes

5. Is the manuscript presented in an intelligible fashion and written in standard English?

Reviewer #1: Yes

Reviewer #3: Yes

Reviewer #1: The authors have thoroughly addressed all previous comments and have substantially improved the manuscript. I recommend it for acceptance and publication.

Reviewer #3: Please use the space provided to explain your answers to the questions above. You may also include additional comments for the author, including concerns about dual publication, research ethics, or publication ethics:

None

**Do you want your identity to be public for this peer review?** For information about this choice, including consent withdrawal, please see our Privacy Policy

Reviewer #1: **Yes: ** Elisabete Teixeira

Reviewer #3: No

---

## [Editor Report · Acceptance letter]

PONE-D-25-10645R1

PLOS ONE

Dear Dr. Suñé Mattevi,

I'm pleased to inform you that your manuscript has been deemed suitable for publication in PLOS ONE. Congratulations! Your manuscript is now being handed over to our production team.

Kind regards,

on behalf of

Dr. Paula Boaventura

Academic Editor

PLOS ONE